# Dissociation between phase and power correlation networks in the human brain is driven by co-occurrent bursts

Rikkert Hindriks [1✉] & Prejaas K. B. Tewarie [2,3]

Well-known haemodynamic resting-state networks are better mirrored in power correlation networks than phase coupling networks in electrophysiological data. However, what do these power correlation networks reflect? We address this long-outstanding question in neuroscience using rigorous mathematical analysis, biophysical simulations with ground truth and application of these mathematical concepts to empirical magnetoencephalography (MEG) data. Our mathematical derivations show that for two non-Gaussian electrophysiological signals, their power correlation depends on their coherence, cokurtosis and conjugate-coherence. Only coherence and cokurtosis contribute to power correlation networks in MEG data, but cokurtosis is less affected by artefactual signal leakage and better mirrors haemodynamic resting-state networks. Simulations and MEG data show that cokurtosis may reflect co-occurrent bursting events. Our findings shed light on the origin of the complementary nature of power correlation networks to phase coupling networks and suggests that the origin of resting-state networks is partly reflected in co-occurrent bursts in neuronal activity.

[1] Department of Mathematics, Faculty of Science, Vrije Universiteit Amsterdam, Amsterdam, The Netherlands. [2] Clinical Neurophysiology Group, University of Twente, Enschede, The Netherlands. [3] Sir Peter Mansfield Imaging Center, School of Physics, University of Nottingham, Nottingham, UK. ✉email: r.hindriks@vu.nl

Cognitive, sensory and motor processing result from synchronous activity between functionally specialised brain regions. Measures for synchrony in electrophysiological brain signals, or also called functional connectivity, can be divided into two broad classes: phase and amplitude coupling[1]. These so-called intrinsic modes of coupling in electrophysiological data are believed to contain common[2,3], but also complementary and distinct information of ongoing neuronal communication[4]. While amplitude coupling captures coordination between slow fluctuations in signal strength, phase coupling is sensitive to synchrony on shorter time-scales[5]. Phase synchrony has historically been the more widely applied class of metric, and its mechanism of action in neuronal communication is fairly well-understood in terms of the "communication through coherence" hypothesis[6]. Recent years have seen a boost in the use of amplitude coupling measures in electrophysiological data[7]. Amplitude coupling comes with higher within-subject consistency between recording sessions and shows clear complementary information during different types of cognitive tasks[8,9]. However, to date, the mechanism of action of amplitude coupling is still poorly understood.

Two brain regions can exhibit amplitude coupling and at the same time have independent phase-dynamics[4,10]. This distinct nature of phase and amplitude coupling becomes eminently apparent in search for the neurophysiological correlate of haemodynamic resting-state networks[11,12]. These resting-state networks reflect spatially coordinated activity in slow haemodynamic signals across distant brain regions, have strong functional and clinical relevance and have become an indispensable element in the field of functional neuroimaging[13]. While resting-state networks can be robustly extracted from electrophysiological recordings using amplitude coupling, these networks are less apparent in the spatial structure of phase coupling networks[12,14–18], leading to the notion that the neurophysiological correlate of resting-state networks should be found in the neuronal mechanism that gives rise to amplitude coupling.

It is not completely understood how differences in phase and amplitude coupling are related to other properties of the signals. Such understanding, however, might be helpful in sorting out the physiological mechanisms that underlie these two types of connectivity and will also guide the development of analysis methods. There are numerous implementations of phase coupling metrics, the phase-locking value, the (weighted) phase-lag index[19,20], coherence and its imaginary part[21], measures based on phase-coupling functions[22], and the lagged coherence[23], among many others[24,25]. Amplitude coupling is usually expressed in instantaneous correlations between (orthogonalized) amplitudes[10,12,15]. In recent work, a relationship between the amplitude correlation and coherence was derived for infinitely long Gaussian signals[26]. A pair of time-domain signals is referred to as *Gaussian* if the pairs of samples are drawn from a bivariate Gaussian distribution. In EEG/MEG studies, signals are usually analysed in the frequency-domain and hence are complex-valued. In this case, a pair of signals is Gaussian if the four-vectors of the real and imaginary parts of the samples are drawn from a four-dimensional Gaussian distribution. Note that, whether or not a pair of signals is Gaussian, does not imply anything about the temporal structure of the signals and, in particular, is unrelated to the signals' auto- and cross-correlation functions.

Nolte and co-workers[26] showed that the amplitude correlation between two signals is equal to the squared magnitude of their coherence. This shows that a proper comparison between coherence and amplitude correlation networks involves the squared magnitude of the coherence, rather than its magnitude, as is sometimes done in practice. Furthermore, and perhaps more importantly, it demonstrates that differences between coherence and amplitude correlation networks can only exist for non-Gaussian signals. For a pair of zero-mean Gaussian signals, all statistical information is contained in their second-order moments, namely the signals' variances and their covariance (time-domain) or cross-spectrum (frequency domain) and the Gaussian distribution is the only distribution with this property. Thus, the observed dissociation between coherence and amplitude correlation networks in EEG and MEG data[4,26] implies that second-order moments do not contain all information about the signals. This additional information, however, can only be obtained by considering higher-order moments of EEG/MEG signals.

There is strong evidence from electroencephalography (EEG) and magnetoencephalography (MEG) recordings that ongoing neuronal oscillations are non-Gaussian[22,27,28], which ever since has a strong impact on the study of neuronal oscillations in electrophysiological data[29]. Non-Gaussianity in neuronal oscillations is expressed as transitions between low- and high-amplitude oscillations[27,28,30]. More specifically, there is a dominant low amplitude mode during spontaneous oscillations, interspersed with brief periods of high amplitude activity, so-called bursts[29,31]. Spontaneous switching between low- and high amplitude oscillations naturally occurs when the system is found near a dynamic instability, more specifically near a subcritical Hopf bifurcation[27]. However, the implications of this dynamical instability for amplitude correlation remain unclear.

An exact mathematical description of amplitude coupling should capture the non-Gaussianity of electrophysiological data. In this study we generalize the relation on amplitude coupling derived by Nolte and co-workers to arbitrary non-Gaussian signals of finite-length using higher-order statistics. We derive an expression for amplitude coupling that is exact and holds for any two non-Gaussian signals. This expression has three important implications: (1) it explains the common and complementary nature of phase and amplitude coupling; (2) it elucidates the contribution of coincident bursting events to amplitude coupling; (3) it hence provides insight into the electrophysiological signal properties that result in amplitude coupling. We extend the derived expression for orthogonalized signals that apply to arbitrary non-Gaussian signals of finite length. We first demonstrate the relevance of our mathematical expression in simulations with ground truth when the system operates on the edge of instability (a so-called subcritical Hopf bifurcation), to probe the role of intermittend bursting in amplitude coupling. We further demonstrate the relevance of our derived expression in empirical MEG data, and show the contribution of coincident bursting events to amplitude coupling with and without leakage correction.

## Results

**Theory: mathematical relationship between power correlation and coherence**. To allow for simplicity in our mathematical derivations, we consider power envelope correlations, rather than amplitude envelope correlations, with the former being defined as the magnitude squared of the amplitude envelopes. We will use coherence as a proxy for phase coupling. Let $x$ and $y$ be the coefficients of frequency representations of two recorded brain signals, for example as obtained by Fourier, wavelet, or Hilbert transformation. In the 'Methods' section we derive the following expression for the power correlation $r_{x,y}$, which can be understood in terms of a relation between the power correlation and the coherence $\rho_{x,y}$ between $x$ and $y$:

$$r_{x,y} = \frac{|\rho_{x,y}|^2 + K_{x,y} + |\rho_{x,\bar{y}}|^2}{\sqrt{(1 + K_x + |\rho_{x,\bar{x}}|^2)(1 + K_y + |\rho_{y,\bar{y}}|^2)}}, \qquad (1)$$

Here, $\bar{x}$ denotes the complex conjugate of $x$, $K_x$ and $K_y$ are the (excess) kurtosis of $x$ and $y$, and $K_{x,y}$ is the (excess) cokurtosis between $x$ and $y$. All quantities in Eq. (1) are estimates of the respective theoretical quantities and can be computed directly from experimental signals. For ongoing signals, they are obtained by averaging over time and for task-based signals, they are obtained by averaging over trials. No assumptions are made in deriving Eq. (1), hence it holds for any two signals of arbitrary length, be it random, non-stationary, chaotic, or otherwise. For later use, we will refer to the coherence $\rho_{x,\bar{y}}$ between $x$ and $\bar{y}$ as the *conjugate coherence* between $x$ and $y$[32]. The conjugate coherence of a signal with itself measures the extent to which the distribution of its instantaneous phase $\arg(x)$ deviates from being uniform. In particular, it vanishes if and only if the phase is uniformly distributed or, equivalently, the real and imaginary parts of the signal have the same variance and are uncorrelated. In a similar way, the conjugate coherence between two signals measures the extent to which the distribution of the sum $\arg(x) + \arg(y)$ deviates from being uniform. Details are provided in the 'Methods' section.

The kurtosis of a signal can be thought of as measuring the "fatness" of the signal's probability distribution, relative to that of a Gaussian reference signal with matched first- and second-order moments (i.e. mean and variance). Thus, signals with positive kurtosis exhibit large fluctuations from their mean with higher probability than matched Gaussian signals do. Signals with positive and negative kurtosis are referred to as *super-Gaussian* and *sub-Gaussian*, respectively. Figure 1a provides an illustration. The cokurtosis between two signals is a measure for the probability of large fluctuations to occur *simultaneously* in both

signals, relative to that of Gaussian reference signals with the same first- and second-order moments (i.e. mean, variance, and coherence). The kurtosis and cokurtosis are fourth-order quantities that vanish for Gaussian signals and, as such, can be used to detect and characterize non-Gaussian signal properties.

For simplicity we now assume that the MEG signals are jointly stationary. This assumption is in no way essential, but simplifies our analysis of Eq. (1), because it implies that the conjugate coherence terms vanish[33]. We will, however, also establish this for the empirical MEG signals in our dataset. Vanishing of the conjugate coherence term implies that the theoretical relation between power correlation and coherence for such signals is

$$r_{x,y} = \frac{|\rho_{x,y}|^2}{\sqrt{(1+K_x)(1+K_y)}} + \frac{K_{x,y}}{\sqrt{(1+K_x)(1+K_y)}}. \quad (2)$$

Equation (2) shows that the power correlation between cortical signals can be decomposed into two parts. The first part reflects dependence on coherence and the synchronization of Gaussian (i.e. non-bursty) background activity. The second part depends on cokurtosis and vanishes for Gaussian signals. We will refer to it as the *non-Gaussian power correlation* and denote it by $r_{x,y}^e$ (the e stands for "excess"):

$$r_{x,y}^e = \frac{K_{x,y}}{\sqrt{(1+K_x)(1+K_y)}}. \quad (3)$$

Equation (3) makes clear that $r_{x,y}^e$ is a natural measure for the dissociation between power correlation and coherence and that this dissociation is closely related to non-Gaussian properties of

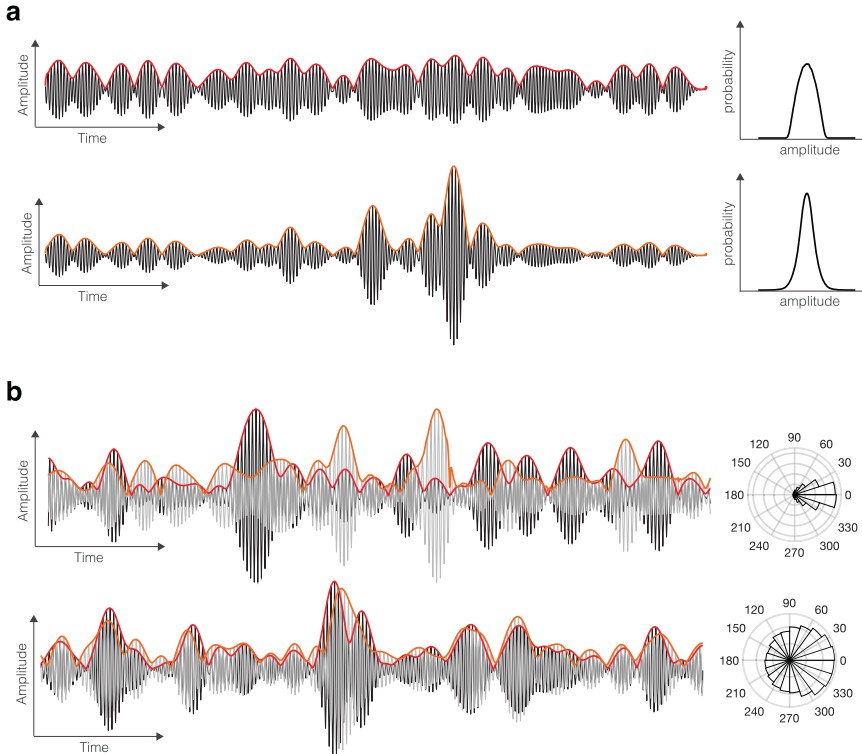

**Fig. 1 Illustration of co)kurtosis of cortical signals. a** Shown are two simulated cortical signals (black traces), together with their amplitude envelopes (red and orange traces) and distributions. The top and bottom signals have negative and positive kurtosis, respectively. Negative kurtosis (sub-Gaussiannity) is reflected in the absence of large amplitude fluctuations and in "thin tails" of the distribution. Positive kurtosis (super-gaussiannity) is reflected in the presence of large amplitude fluctuations and in "fat tails" of the distribution. **b** Top: Shown are two simulated super-Gaussian cortical signals (black and grey traces), together with their amplitude envelopes (red and orange traces, resp.) and the distribution of the difference between their instantaneous phases. The magnitude squared coherence is larger than the correlation between their power envelopes, corresponding to a negative cokurtosis. Bottom: Same format, but the magnitude squared coherence is smaller than the power correlation, corresponding to a positive cokurtosis.

the signals. For instance, it makes clear that a dissociation between power correlation and coherence is only possible for non-Gaussian signals, something that has not been recognized in earlier work, but will play a crucial role in obtaining a physiological explanation of this dissociation.

How should a positive non-Gaussian power correlation be interpreted in terms of the dynamics of the neural populations that generate the signals? We first note that cortical signals are super-Gaussian[22,27]. For such signals, Eq. (3) implies that a sufficient condition for the non-Gaussian power correlation to be positive is that $r_{x,y} > |\rho_{x,y}|^2$. A positive value of $r_{x,y}$ therefore means that the signals are relatively incoherent as compared to their power correlation. In particular, incoherent signals with correlated power fluctuations have positive non-Gaussian power correlation. Figure 1b provides an illustration. The non-Gaussian power correlation captures the occurence of simultaneous large amplitude fluctuations in two signals. Since these large amplitude fluctuations correspond to the tails in the distribution, their probability of occurence is low. Hence, a plausible neurophysiological explanation for a positive non-Gaussian power correlation is the coincidence of high amplitude bursts. We test this hypothesis using biophysical modelling based simulations with ground truth.

**Corticothalamic mean-field simulations: non-Gaussian power correlations reflect synchronized neuronal bursts**. We employed the same corticothalamic mean-field model as Freyer and coworkers[27,34]. This model simulates electrophysiological signals generated by an excitatory pyramidal neuronal population in terms of its mean firing rate $\phi_e$ (Fig. 2a). The model produces 1/f activity superimposed by alpha oscillations, which emerge from a thalamocortical loop, consisting of two thalamic (relay and reticular) and two cortical (excitatory and inhibitory) populations. Working point of the model is near a dynamical instability, a subcritical Hopf bifurcation. In the presence of noise, this bifurcation separates a linear regime, characterised by noise induced low amplitude fluctuations and a limit cycle regime, characterised by high amplitude oscillations (Fig. 2b). Similar as in the work of Freyer and coworkers[27], switching between these regimes is controlled by state-dependent noise parameter $\chi$, which results in alternating modes of low amplitude fluctuations and brief high amplitude bursts. To model coincident bursting, we couple the excitatory populations of two corticothalamic mean-field models, controlled by the coupling strength.

Electrophysiological signals were generated using a range of values for $\chi$ and coupling strength, starting both at zero magnitude. For $\chi = 0$ (no bursting events), we observe increasing power correlation values when coupling strength is increased, resulting from increasing values for coherence (Fig. 2c). Cokurtosis remains zero for this parameter range. For coupling > 0.06 and for increasing values of $\chi$ (or bursting events), distinct model behaviour is observed. For these parameter values, there is a boost in magnitude of the power correlation due to a jump in cokurtosis, but vanishing coherence (Fig. 2c). Hence, coincident bursting is well captured by cokurtosis, but not by coherence. Figure 2c clearly shows two regimes responsible for non-zero power correlation, a high coupling and non-bursting regime characterised by non-zero coherence, and a high coupling and bursting regime characterised by positive cokurtosis.

**Empirical MEG data: dissociation between cortical coherence and power correlation networks**. After establishing an interpretation for the derived mathematical expression of Eq. (1) using simulations, we set out to analyse the role of non-Gaussian power correlations and cokurtosis in empirical MEG data. Equation (1) implies that for Gaussian signals, the theoretical power correlation equals the squared magnitude coherence. To assess if cortical signals are indeed non-Gaussian and how the extent of non-Gaussianity varies with cortical location, we calculated the kurtosis for all cortical regions. The results are reported in Supplementary Figs. 1 and 2 of Supplementary Note 1 and demonstrate that cortical signals are super-Gaussian.

Given this observation, we expect to see a dissociation between the squared magnitude coherence and the power correlation. We estimated the power correlation and magnitude squared coherence between the cortical signals for all pairs of regions and averaged their differences over subjects. Figure 3a shows the distributions of this difference in the alpha and beta bands, which were obtained by pooling the values from all pairs of regions. To assess statistical significance, we recomputed the differences for randomized MEG signals. The obtained null-distributions are shown in Fig. 3a (black and grey curves). Their 95% percentiles are 0.007 (alpha band) and 0.006 (beta band), which shows that the power correlation and coherence are dissociated for most

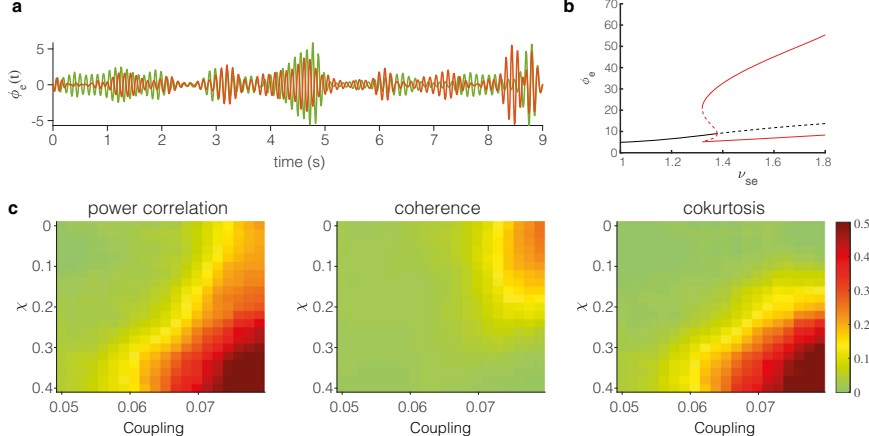

**Fig. 2 Coincident bursts in a corticothalamic mean-field model.** We simulated activity of two coupled corticothalamic mean-field models. An example of two generated electrophysiological signals is depicted in panel (**a**). Panel (**b**) shows the corresponding bifurcation diagram of the model, when the thalamocortical connection strength $\nu_{es}$ is used as bifurcation parameter. The black line corresponds to the low amplitude mode (linear regime) and the red lines to the high amplitude mode (limit cycle regime). State-dependent noise, controlled by parameter $\chi$ allows for switching between these two regimes. Panel (**c**) shows the power correlation, coherence and cokurtosis as a function of $\chi$ and coupling strength.

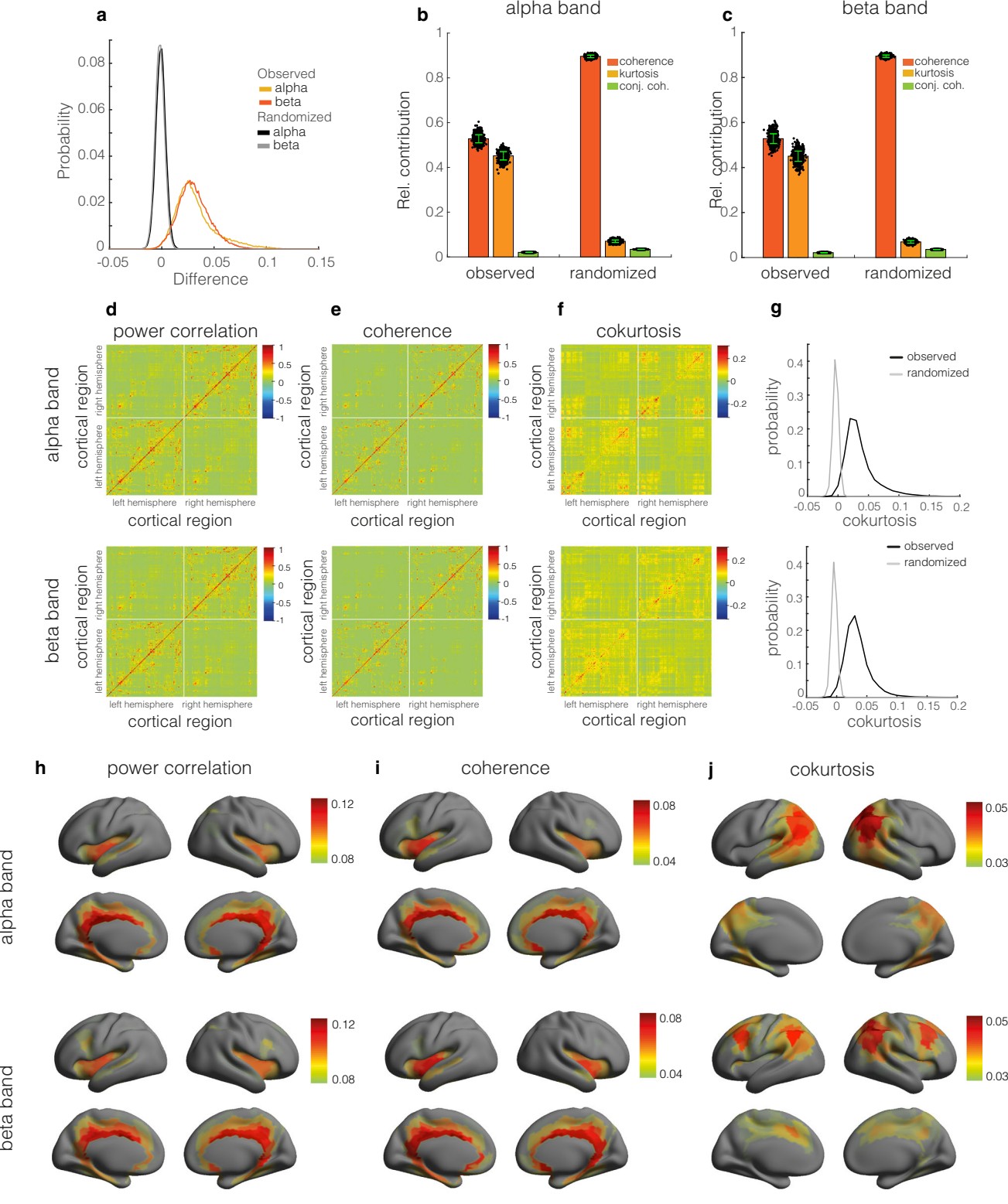

pairs of regions and, more specifically, that power correlations are consistently higher than expected for Gaussian signals, corroborating recent findings using EEG sensor data[26]. Since the difference is always positive, it also shows that coherent cortical oscillations necessarily have correlated power fluctuations, in line with recent observations[4].

We next quantified the contributions of the coherence, cokurtosis, and conjugate coherence to the power correlations. This was done by first estimating their contributions, corresponding to the three terms in Eq. (1), for all pairs of regions, summing them over all pairs of regions, and dividing by the total sum. The results are shown in Fig. 3b, c for the alpha and beta bands, respectively. In both frequency bands, the contributions of the coherence and cokurtosis are roughly equal and the contribution of the conjugate coherence is very small. The figures also show the contributions obtained from randomized MEG signals. The randomization caused the contribution of the cokurtosis to decreased substantially. Since the randomized

**Fig. 3 Dissociation between cortical coherence and power correlation networks. a** Distributions of the observed differences (power correlation minus squared magnitude coherence) in the alpha (light orange curve) and beta (dark orange curve) frequency bands and the corresponding null-distributions in black (alpha band) and grey (beta band). The distributions were obtained by averaging the differences over subjects and subsequently pooling the values from all pairs of cortical regions. **b** Relative contributions of coherence (dark orange bars), cokurtosis (light orange bars), and conjugate coherence (green bars) to cortical power correlations in the alpha frequency band. The contributions were averaged over pairs of cortical regions. The contributions obtained from a randomized copy of the MEG sensor signals are also shown. **c** Same format as (**b**) but for the beta frequency band. The black dots in panels (**b**) and (**c**) quantify the statistical uncertainty of the relative contributions and were obtained by bootstrapping over subjects (1000 bootstrap realizations). The corresponding standard errors are also shown (green vertical bars). **d** Power correlation network in the alpha (top panel) and beta (bottom panel) frequency band. **e** Coherence network in the alpha(top panel) and beta (bottom panel) frequency band. **f** Cokurtosis network in the alpha (top panel) and beta (bottom panel)frequency band. The networks were obtained by averaging the respective subject-specific network matrices. **g** Distribution of the cokurtosis values in the alpha (top panel) and beta (bottom panel) frequency band (black curve) and of those of a randomized copy of the MEG signals (grey curve). **h** Colour-coded cortical map of the region-averaged power correlations (i.e. the row-averaged power correlation matrix) in the alpha (top panel) and beta (bottom panel) frequency band. **i** Colour-coded cortical map of the region-averaged coherence (i.e. the row-averaged coherence matrix) in the alpha (top panel) and beta (bottom panel) frequency band. **j** Colour-coded cortical map of the region-averaged cokurtosis (i.e. the row-averaged cokurtosis matrix) in the alpha (top panel) and beta (bottom panel) frequency band. For better visibility, the cortical maps were thresholded at their average values.

signals are Gaussian and hence have zero cokurtosis, the contributions of the kurtosis to the power correlations of the randomized signals is entirely due to sampling variability. This demonstrates that the contribution of the kurtosis to the power correlation is not spurious and is not caused by the processing and inverse modelling of the MEG signals. Furthermore, since the results are highly reproducible across recording sessions, the sampling variability of the estimated contributions is very small. These findings establish that the dissociation of power correlation and coherence in cortical oscillations reflects extreme co-fluctuations in the oscillations' power envelopes, as measured by the cokurtosis, and simulations suggest that this could be mirrored by synchronised bursts.

By collecting the contributions of the coherence, cokurtosis, and conjugate coherence into matrices, an additive decomposition of the observed power correlation matrix into three matrices is obtained. Figure 3d shows the power correlation networks and Fig. 3e, f shows the corresponding coherence and co-kurtosis networks. Because the entries of the conjugate coherence matrix are extremely small (at least an order of magnitude smaller than those of the other matrices), the power correlation matrix is, to a very good approximation, the sum of the coherence and the cokurtosis matrices. Figure 3f shows that the cokurtosis is non-negative for all pairs of regions and for most pairs is strictly positive. This can also be observed in Fig. 3g, which show the distributions of the matrix entries (black curves) and the corresponding null-distributions (grey curves). We further observe that the cokurtosis matrices contain more off-diagonal elements and hence less connections between adjacent regions than coherence matrices. There are a large number of region-pairs that are incoherent, but have non-zero and positive cokurtosis.

To see the dependence on cortical location, we calculated seed-based maps by averaging over the rows of the cukurtosis, coherence and power correlation matrices (Fig. 3h–j). In both frequency bands, the seed-based power correlation and coherence are relatively high in deep cortial regions, such as the Sylvian fissures and along the cingular cortices and are most likely artificial[12]. The seed-based cokurtosis is organized differently. In the alpha band, it is high in parietal cortices, including the precuneus, and the posterior cingular cortices and in the beta band it is high in the regions of the well-known fronto-parietal attention network. This spatial organization is highly similar to that observed in ref. [12] using amplitude correlations of orthogonalized MEG signals. Orthogonalization is a processing step that is used to suppress spurious power correlations that are caused by incomplete unmixing of the reconstructed cortical signals. Our results suggest that this step is not necessary when

using only the non-Gaussian part of the power correlation (Eq. (3)). This is confirmed in the section *Effect of signal orthogonalization*. All observations were highly reproducible in a separate recording session (see Supplementary Fig. 3).

**Empirical MEG data: resting-state networks are reflected in non-Gaussian power correlations.** To extract non-Gaussian power correlation subnetworks, we clustered the columns of the subject-averaged cokurtosis matrices using *k*-means clustering. The number of clusters was determined by the elbow method applied to the average sum of squared within-cluster distances to the cluster centers. In selecting the optimal number of clusters, the number of clusters was allowed to range from one to ten. For each of the ten values, we selected the best cluster out of ten replications with different initial conditions. Figure 4a shows the subnetworks extracted in the alpha frequency band. The first network coincides with the visual network and covers both dorsal and ventral visual regions. The second network covers the posterior cingular cortex and parietal areas, including the precuneus, which is the posterior node of the default mode network (DMN). The third network comprises the medial frontal, temporal, and parietal areas of the DMN as well as the auditory network. When the number of clusters is increased from three to four, the auditory networks separates from the DMN. Figure 4b shows the networks obtained in the beta frequency band. The first network is the visual network, and the second an amalgam of the sensorimotor network with more parietal regions. The third network comprises areas in the dorsal and medial pre-frontal cortices and shares similarities with the salience network. Applying the same procedure to the MEG data from the second recording session yielded identical networks (see Supplementary Fig. 4). These results demonstrate that some of the resting-state networks, in particular the DMN and the salience network, can be extracted from MEG signals using non-Gaussian power correlations and without the need for signal orthogonalization.

**Empirical MEG data: effect of signal orthogonalization.** When using power correlations for the reconstruction of cortical networks, the signals need to be orthogonalized prior to estimating the correlations[12]. This is to remove spurious correlations caused by signal leakage, i.e. incomplete unmixing of the reconstructed cortical signals. To assess the effects of signal orthogonalization on the coherence and cokurtosis networks, we calculated the respective network matrices obtained from the orthogonalized signals. Figure 5a, b shows the relative contributions of coherence, cokurtosis, and conjugate coherence to the power correlation. Comparing the contributions to those obtained without

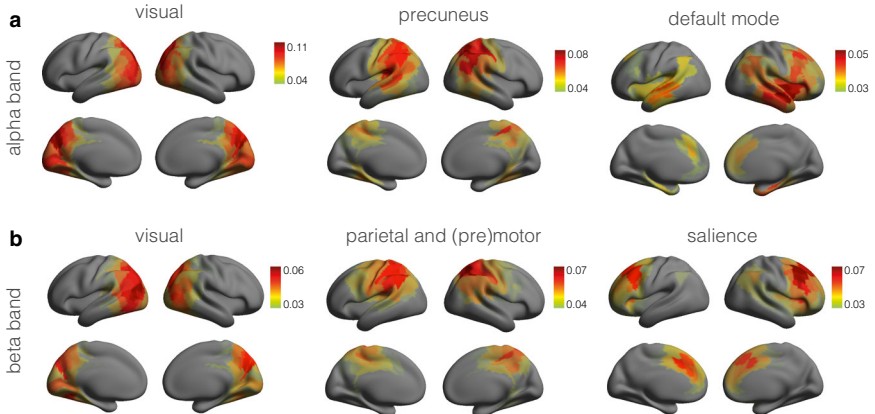

**Fig. 4 Non-Gaussian power correlation networks. a** Non-Gaussian power correlation networks extracted from spontaneous cortical oscillations in the alpha frequency band. The ordering of the networks is arbitrary. **b** Non-Gaussian power correlation networks extracted from spontaneous cortical oscillations in the beta frequency band. The networks in both panels were extracted by clustering of the group-level seed-based cokurtosis matrices in the respective frequency bands. Displayed are the cluster centers. For better visibility, the networks were thresholded at their average values.

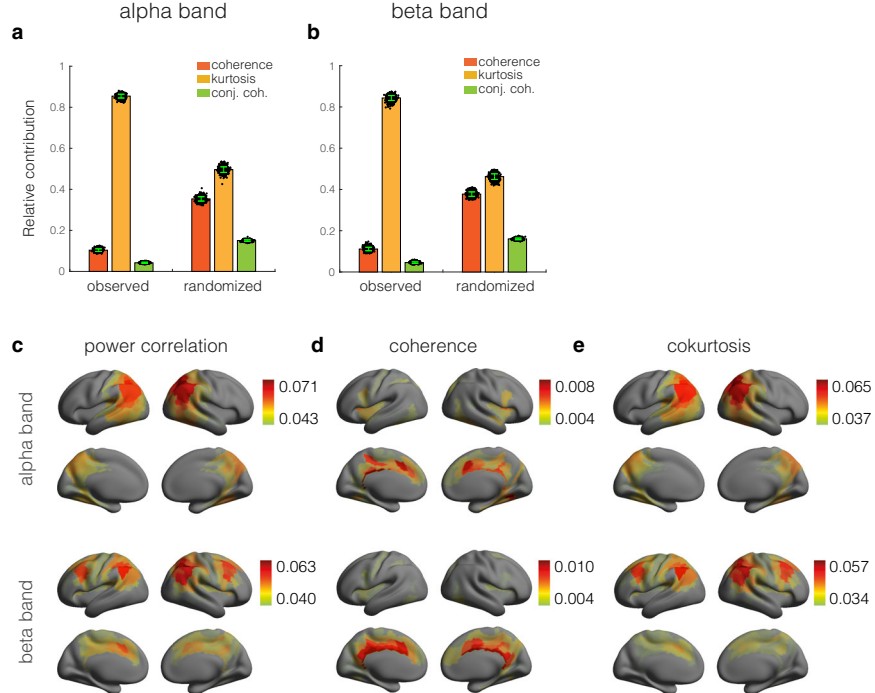

**Fig. 5 Effect of signal orthogonalization. a** Relative average contributions of coherence (dark orange bars), cokurtosis (light orange bars), and conjugate coherence (green bars) to cortical power correlations in the alpha frequency band. The contributions obtained from a randomized copy of the MEG sensor signals are also shown. **b** Same format as (**a**) but for the beta frequency band. The black dots in panels (**a**) and (**b**) quantify the statistical uncertainty of the relative contributions and were obtained by bootstrapping over subjects (1000 bootstrap realizations). The corresponding standard errors are also shown (green vertical bars). **c** Colour-coded cortical maps of the region-averaged power correlation network (i.e. the row-averaged power correlation matrix) in the alpha (top) and beta (bottom) frequency bands. **d** Colour-coded cortical maps of the region-averaged coherence network (i.e. the row-averaged power coherence matrix) in the alpha (top) and beta (bottom) frequency bands. **e** Colour-coded cortical maps of the region-averaged cokurtosis network (i.e. the row-averaged power cokurtosis matrix) in the alpha (top) and beta (bottom) frequency bands. For better visibility, the cortical maps were thresholded at their average values.

orthogonalization (Fig. 3b, c) makes clear that the contribution of the conjugate coherence remains small and that the contribution of the coherence is reduced from about 50% to less than 10%. This shows that the power correlation between orthogonalized signals roughly measures their non-Gaussian power correlation. It also explains why signal orthogonalization is not necessary when using the cokurtosis to map cortical connectivity. Indeed, the spatial correlation between the power correlation and cokurtosis matrices of the orthogonalized signals are 0.99 and

0.98 for the alpha and beta bands, respectively. In the 'Methods' section we provide a mathematical analysis of these effects.

Figure 5c–e shows the seed-based power correlation, coherence, and cokurtosis maps. The power correlation maps show that the orthogonalization has removed a large part of the spurious connectivity in both frequency bands (compare with the maps in Fig. 3h–j). However, the coherence maps, which seem to mostly represent spurious connectivity in deep cortical regions such as the cingulate gyri and the insular cortex, still contribute to

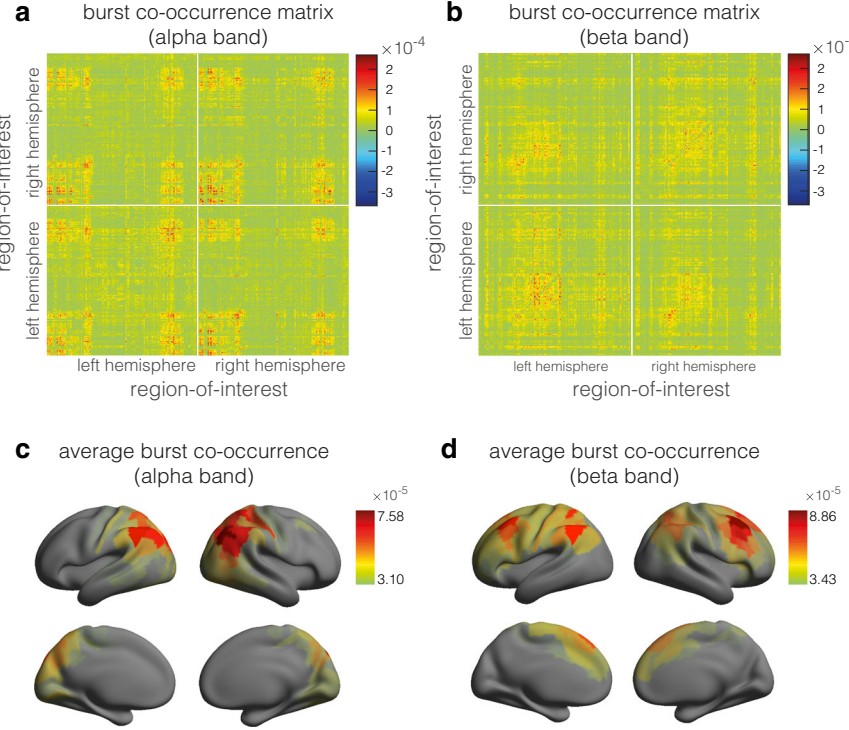

**Fig. 6 Burst co-occurrences in resting-state cortical activity. a** Burst co-occurrence matrix in the alpha frequency band. **b** Burst co-occurrence matrix in the beta frequency band. **c** Colour-coded cortical map of the region-averaged burst co-occurrence in the alpha frequency band. **d** Colour-coded cortical map of the region-averaged burst co-occurrence in the beta frequency band. The matrices and maps were obtained by averaging over subjects. For better visibility, the cortical maps were thresholded at their average values.

the power correlations. Power correlations between orthogonalized signals, therefore, are not completely free of spurious connectivity caused by signal leakage. This is consistent with the fact that orthogonalization completely suppresses signal leakage only for Gaussian signals[26]. In contrast, the cokurtosis maps are hardly affected by the orthogonalization. Indeed, the spatial correlation between the cokurtosis matrices of the original and orthogonalized signals is 0.93 and 0.91 for the alpha and beta bands, respectively. From this we conclude that signal leakage strongly affects Gaussian power correlations, but not non-Gaussian ones, although the cokurtosis maps are slightly altered by orthogonalization (see Supplementary Fig. 5). All of these results were highly reproducible in a separate recording session (see Supplementary Fig. 6).

**Empirical MEG data: non-Gaussian power correlations reflect co-occurrent bursts in cortical activity.** To relate the cokurtosis networks to more familiar properties of cortical signals, we calculated the group-level burst co-occurrence matrices of orthogonalized source-projected MEG signals in the alpha and beta frequency bands. Like the cokurtosis, the burst co-occurrence between two signals is a measure for the probability of the signals to simultaneously exhibit large power fluctuations. The burst occurrence, however, is more in line with analysis methods of resting-state cortical activity that are based on point-processes (see e.g. ref. [35]). Because the analysis is carried out in the time-domain, our analysis also serves to illustrate the application of non-Gaussian power correlations in the time-domain.

The burst co-occurrence matrices are shown in Fig. 6a (alpha band) and b (beta band). They resemble the cokurtosis matrices extracted in the frequency domain: The spatial correlations between cokurtosis and burst co-occurrence matrices are 0.58 (alpha band) and 0.48 (beta band). Figure 6c, d shows the burst

co-occurrence matrices averaged over regions-of-interest. The maps clearly resemble the cokurtosis maps (Fig. 3f) in both frequency bands. In particular, the fronto-parietal attention network can be observed in the beta band. Lastly, we computed the cokurtosis matrices from the time-domain sensor signals and compared them with those obtained from the frequency-domain sensor signals. The correlations between the respective matrices were 0.96 in both frequency bands, showing that non-Gaussian power correlations can be extracted in both the time- and frequency domain and that the results agree.

## Discussion

What do power correlation networks in the human brain reflect and how do they relate to coherence networks? This has been a long-outstanding question in the field of neuroscience and relevant for several neuroscientific experiments using modalities such as local field potentials, electrocorticography, EEG and MEG. We provide an answer to this outstanding question by deriving an analytical mathematical expression for the power correlation between two non-Gaussian electrophysiological signals and dissecting this further using biophysical modelling and empirical MEG data. Both coherence and cokurtosis contribute to a power correlation between two non-Gaussian electrophysiological signals. While coherence is sensitive to phase synchrony, cokurtosis or its normalised version, the non-Gaussian power correlation, captures the probability of simultaneously occurring large fluctuations in two electrophysiological signals. Simulations showed that cokurtosis is especially sensitive to coincident bursts. This has been confirmed in empirical MEG data, showing a strong relation between cokurtosis and coincident bursts for both alpha and beta networks. Empirical MEG data further showed a clear dissociation between coherence and power correlation for alpha and beta networks, which was explained by a positive cokurtosis.

Positive cokurtosis was observed for several long-distance region-pairs for which there was near zero coherence. Well-known resting-state networks were clearly apparent in cokurtosis without the need for correction for artefactual signal leakage. Correction for artefactual signal leakage had profound effect on coherence, but not on cokurtosis, indicating that cokurtosis is relatively insensitive to signal leakage.

The origin of power correlations for orthogonalised electrophysiological signals seems consistent with coincident bursting in two separate brain regions. As (positive) kurtosis corresponds to extreme values within the tails of the signal's distribution. Their probability of occurrence is low and duration of large fluctuations is expected to be brief. Sustained oscillations with high amplitude do not seem to agree with current findings as this would likely result in negative kurtosis. Furthermore, there is increasing evidence in that bursting events may even be the underlying features of the alpha and beta rhythm[36–40]. Thus, other explanations for the observed positive kurtosis and cokurtosis than bursting events seem unlikely. We identified bursting events using a point process analysis[35,41], which indeed shows that coincident bursts were highly correlated to cokurtosis. The strength of the current analysis is that this very simple method already demonstrates the relationship between coincident bursting events and power correlations. However, more sophisticated analysis tools exist to capture bursting events, such as the Hidden Markov modelling approach that is sensitive to distinct spectral features of bursts[31] and the "better oscillation detection" (BOSC) method, which are presumably less sensitive to noise and artefacts in the data[42]. The role for coincident bursting may be supported by studies on time-resolved functional connectivity that have demonstrated very brief periods (~0–400 ms) at which dynamic amplitude connectivity exceeds the level of stationary connectivity[43–45]. Importantly, a previous paper on coincident bursting using Hidden Markov Modelling indeed showed striking resemblance of coincident bursting events to power correlations[46], though this was only analysed by assessing correlations without offering a clear mathematical and explanatory framework that explains the dissociation between coherence and power correlations. Recent years have seen a boost in the interest of bursting activity in neuronal activity[29,31,47]. Bursts can be dissected into burst amplitude, duration and occurrence[31]. Future work will need to examine how these bursting properties affect coincident bursting and subsequently cokurtosis.

Using biophysical simulations, we demonstrated that a system in the vicinity of a dynamical instability is not only in agreement with switching between low amplitude fluctuations and high amplitude bursts[27], but also in agreement with occurrence of coincident bursts. More specifically, previous work has demonstrated that state-dependent noise in a system operating near a subcritical Hopf bifurcation can cause erratic jumps between low amplitude fluctuations and high amplitude bursts, agreeing with empirical observations of the alpha rhythm[27]. When two coupled neuronal populations both operate at the brink of this instability, a brief burst in amplitude in one population can make the other population jump from the low to high amplitude regime, thus leading to coincident bursting. The agreement of a system characterised by a Hopf bifurcation and empirical observations is not novel. The relevance of a Hopf bifurcation has been described in the context of several other empirical observations, such as frequency-dependent electrophysiological activity[48], temporal evolution of the oscillatory amplitudes[49], turbulent dynamics in the human brain[50], and transitions between brain states[51,52]. As such, the potential role of a Hopf bifurcation in the context of coincident bursting can now be added to this list. On a different note, to be a realistic mimic for co-occurent bursting in empirical electrophysiological data, (biophysical) simulations should exhibit

positive kurtosis and cokurtosis, which can be influenced by the choice of the coupling function, e.g. diffusive or additive[53]. Diffusive coupling may lead to a reduction in amplitude and hence negative cokurtosis, which may not be appropriate to capture this specific empirical phenomenon of co-occurent bursting.

An important observation is the eminent appearance of well-known haemodynamic resting-state networks in cokurtosis in the alpha and beta networks. Therefore, it is tempting to speculate that these resting-state networks could be shaped by coincident neuronal bursting events. This has also been stressed in animal work aiming to explain spontaneous fMRI correlations[54]. Even though it has been claimed that not more than 10% of the variance of fMRI BOLD can be explained by underlying neuronal events and the contribution of non-neuronal events is inhomogeneous across the brain[55,56], recent work in mice suggest that kurtosis of calcium signals mirrors kurtosis in BOLD signals and fMRI BOLD correlations are related to nonstationarity of the calcium signals. Furthermore, there is increasing evidence that co-occurence of high LFP events is one of the most important contributors to BOLD correlations[57–59]. In our work, even without the use of any clustering algorithms, the salience network could clearly be observed in beta cokurtosis networks. Though spectral power in the alpha band is more occipitally located, cokurtosis for alpha networks showed predominantly parietal regions, including the precuneus. This is in agreement with recent work on the generation and propagation of the alpha rhythm[60]. Spatially most dominant generators of alpha activity are located at parietal regions and alpha activity propagates from parietal to occipital regions[60]. It is tempting to speculate that spatially dominant generators of the alpha rhythm could have large (co) kurtosis and result from bursting events. Recent work on bursting events also showed that bursting events propagate spatially across the cortex in a medial to lateral and anterior-posterior direction[61]. After application of k-means clustering, clear resting-state networks can be observed in both beta and alpha frequency bands, including the sensorimotor, visual, and default mode network. We note that, besides being very similar to the networks obtained by orthogonalized power correlations, these networks are also very similar to the networks obtained using a time-delay embedded hidden Markov model in MEG data[62]. This can be explained by our hypothesis, since the spatial organization of coincident bursting will be imprinted on both the second- and fourth-order statistical structure of the MEG signals. Whereas orthogonalized power correlations and the hidden Markov model capture the second-order (i.e. Gaussian) structure, the cokurtosis captures the fourth-order (i.e. non-Gaussian) structure. The main advantage of using the cokurtosis is that no correction for signal leakage is needed. Furthermore, these resting-state networks were not evident in coherence networks and unlike coherence, cokurtosis networks contained less connections in the midline reminscent of artefactual signal leakage.

Taken together, we propose to use the cokurtosis, or its normalised version, the non-Gaussian power correlation, as a functional connectivity metric. This metric is sensitive only to co-occurring extreme fluctuations and, as such, is well-suited to detect synchronized bursting. Furthermore, it is able to detect well-known resting-state networks without the need for signal orthogonalization. The non-Gaussian power correlation is to be preferred over the cokurtosis, because it is appropriately normalised. More specifically, in the 'Methods' section we derive that the cokurtosis is equal to the "excess" covariance between the power fluctuations in the signals, that is, where "excess" means relative to the covariance between the power fluctuations of matched Gaussian signals. This interpretation makes clear that a properly normalized connectivity measure is obtained by dividing the cokurtosis by the product of the standard deviations of the

power fluctuations, in a similar way as the correlation between two signals is obtained by dividing the covariance by the product of the standard deviations of the signals. It turns out that normalizing the cokurtosis in this way precisely yields the non-Gaussian power correlation. Current findings also have implications for the analysis of time-varying connectivity in electrophysiological data. Time-varying connectivity is only relevant if a system exhibits non-stationarity. Stationary connectivity with noise fluctuations around a steady state can be characterised by a Gaussian distribution. Our method allows to study deviations from this Gaussian distribution and is hence sensitive to non-stationarity. Recently introduced methods on time-varying connectivity making use of co-modulations in high amplitude therefore seem justified given their sensitivity to this non-Gaussianity[63,64]. Lastly, we derived an expression for power correlations in terms of coherence and not directly in terms of phase-locking as measured by the phase-locking value (PLV). Such a relation only exists if assumptions about the signals are made, because the power envelopes and instantaneous phases of a general pair of signals can be specified independently from each other. Nolte and coworkers demonstrated how the PLV can be expressed in terms of coherence for bivariate Gaussian signals[26]. This relationship, however, is quite complicated and involves infinite power series in the coherence. Deriving similar relations for more general (non-Gaussian) models is therefore a challenging question for future studies.

Recent years has seen an increase in interest on local bursting events with claims that neuronal oscillations may actually be a reflection of bursting events[29]. Our work forms a strong link between the field of local bursting events and brain networks. We have provided understanding on the dissociation between power correlation and coherence networks. Their dissociation is driven by cokurtosis and consistent with co-occurrent bursting events. Co-occurrence of neuronal bursts revealed canonical haemodynamic resting-state networks. We have presented a robust method, the non-Gaussian power correlation, that can pave the way for analysis of coincident bursts in the context of neuroscientific experiments. Importantly, current results may guide the further search for neuronal underpinnings of haemodynamic resting-state networks.

## Methods

**Data and pre-processing**. We used MEG data from the Human Connectome project[65] (MEG2 Data Release). It comprised three six-minute resting-state scanning sessions from 89 young healthy participants using the whole-head MAGNUS 3600 (4D Neuroimaging, San Diego, CA) system, housed in a magnetically shielded room. The participants were instructed to remain still (supine position), to relax with eyes open, and to fixate on a projected cross-hair on a dark background. The recorded data were segmented into two-second epochs and pre-processed using a dedicated pipeline that included detection of bad channels and segments and ICA-based artefact rejection (see ref. [66] for details). For the spectral domain analyses, the signals of all $k$ MEG sensors were transformed to the time-frequency domain using the short-time Fourier transform with windows of length one second and fifty percent overlap. Subsequently, the time-frequency coefficients at the group-level alpha (≈10 Hz) and beta (≈16 Hz) peak-frequencies were selected. For the time-domain analyses, the signals were filtered ±2 Hz about the alpha and beta peak-frequencies using a zero-phase fourth-order Butterworth bandpass filter. Analytic signals of the filtered signals were computed using the Hilbert transformation, followed by down-sampling with a factor of five. Both methods yielded a complex MEG sensor matrix $X$ that contained the complex signal representations (either Fourier- of Hilbert-based) in its rows.

**Source modelling**. MEG leadfield matrices were computed in Fieldtrip[67] using realistic single-shell headmodels provided by the HCP. As source spaces we used co-registered individual cortical meshes provided by HCP. A detailed description of the anatomical processing pipeline can be found in ref. [68]. This yielded subject- and run-specific leadfield matrices $G$ of dimension $k \times 3p$, where $k$ is the number of MEG channels and $p = 8004$ is the number of cortical vertices. Columns $3j - 2$, $3j - 1$, and $3j$ of $G$ contain the vertex-wise leadfield vectors at the cortical vertex $j$ for the three Euclidean orientations. We denote this $3 \times k$ matrix by $L_j$. The sensor-level Fourier coefficients were projected to source space by an

adaptive spatial filter[12,69]. The source signals $Y_j$ at vertex $j$ are obtained through

$$Y_j = W_j X,$$

where $W_j$ is the $3 \times k$ matrix of filter weights, $X$ is the MEG sensor matrix, and $Y_j$ is the $3 \times p$ matrix containing the source signals for the three Euclidean orientations. The weight matrix is obtained by solving the following optimization problem:

$$\min_W W_j \Sigma W^\dagger \quad \text{subject to} \quad W^\dagger L_j = I,$$

where $\Sigma = XX^\dagger$ is the sensor covariance and $I$ is the $3 \times 3$ identity matrix. This minimizes the source power under the constraint that the power at vertex $j$ is passed with unit gain. It has the effect that interfering signals from other locations are suppressed. The solution to this optimization problem is

$$W_j = \left( L_j^T \Sigma^{-1} L_j \right)^{-1} L^T \Sigma^{-1},$$

where the inverse sensor covariance matrix is estimated through

$$\Sigma^{-1} = (\Sigma + \lambda I)^{-1},$$

where $\lambda$ is a regularization parameter, which was set to $10^{-7}$. Univariate source time-series were subsequently obtained by projecting $Y_j$ onto the first eigenvector of $W_j \Sigma W_j^\dagger$[12]. The above procedure was repeated for each region-of-interest from the cortical parcellation[70].

**Coherence and power correlation**. Let $x \in \mathbb{C}$ and $y \in \mathbb{C}$ denote the time-frequency coefficients of two signals at a given frequency and let $\langle x \rangle$ and $\langle y \rangle$ be their temporal averages over some observation interval. More generally, we use the notation $\langle f(x, y) \rangle$ to denote the temporal average of the expression inside the brackets. To simplify the formulas, we assume throughout that the signal means $\langle x \rangle$ and $\langle y \rangle$ have been subtracted. The coherence and power correlation between $x$ and $y$ are defined as

$$\rho_{x,y} = \frac{\langle x\bar{y} \rangle}{\sqrt{\langle |x|^2 \rangle \langle |y|^2 \rangle}}, \tag{4}$$

and

$$r_{x,y} = \frac{\langle (|x|^2 - \langle |x|^2 \rangle)(|y|^2 - \langle |y|^2 \rangle) \rangle}{\sqrt{\langle (|x|^2 - \langle |x|^2 \rangle)^2 \rangle \langle (|y|^2 - \langle |y|^2 \rangle)^2 \rangle}}, \tag{5}$$

respectively, where the vertical bars denote taking the absolute value. The coherence is complex and $|\rho_{x,y}| \leq 1$ and the power correlation is real and confined to the interval $[-1, 1]$.

**Kurtosis and cokurtosis**. Let $x_1, x_2, x_3, x_4$ be zero-mean complex random variables. Their fourth-order joint moment is defined as their average of their product:

$$\mu(x_1, \ldots, x_d) = \langle x_1 \ldots x_d \rangle. \tag{6}$$

Note that $\mu$ linear in each of its arguments and is symmetric, i.e. invariant under permutations of its arguments. The fourth-order joint cumulant of $x_1, x_2, x_3, x_4$ is defined as

$$\begin{aligned}\kappa(x_1, x_2, x_3, x_4) = &\mu(x_1, x_2, x_3, x_4) - \mu(x_1, x_2)\mu(x_3, x_4) \\ &- \mu(x_1, x_3)\mu(x_2, x_4) - \mu(x_1, x_4)\mu(x_2, x_3),\end{aligned} \tag{7}$$

where $\mu(x_1, x_2)$ denotes the second-order joint moment of $x_1$ and $x_2$, and similarly for the other terms. The joint cumulant is also symmetric and linear in each of its arguments. The joint moments and cumulants can be normalized to make them dimensionless:

$$\tilde{\mu}(x_1, x_2, x_3, x_4) = \frac{\mu(x_1, x_2, x_3, x_4)}{\sqrt{\langle |x_1|^2 \rangle \langle |x_2|^2 \rangle \langle |x_3|^2 \rangle \langle |x_4|^2 \rangle}}, \tag{8}$$

and

$$\tilde{\kappa}(x_1, x_2, x_3, x_4) = \frac{\kappa(x_1, x_2, x_3, x_4)}{\sqrt{\langle |x_1|^2 \rangle \langle |x_2|^2 \rangle \langle |x_3|^2 \rangle \langle |x_4|^2 \rangle}}. \tag{9}$$

Let $x$ and $y$ be zero-mean complex random variables. There are three normalized fourth-order cumulants that appear in the relation between the coherence and the power correlation between $x$ and $y$ (Eq. (1)). The first two are the kurtosis of $x$ and $y$, which are defined as

$$K_x = \tilde{\kappa}(x, x, \bar{x}, \bar{x}), \tag{10}$$

and

$$K_y = \tilde{\kappa}(y, y, \bar{y}, \bar{y}), \tag{11}$$

respectively, and the third is the cokurtosis between $x$ and $y$, which is defined as

$$K_{xy} = \tilde{\kappa}(x, y, \bar{x}, \bar{y}). \tag{12}$$

Note that $K_x$, $K_y$, and $K_{xy}$ are real-valued.

**Conjugate coherence**. Let $z = (x, y)$ be a zero-mean bivariate complex random variable. Generally, its cross-spectral matrix

$$\langle zz^\dagger \rangle = \begin{pmatrix} \langle |x|^2 \rangle & \langle x\bar{y} \rangle \\ \langle y\bar{x} \rangle & \langle |y|^2 \rangle \end{pmatrix}. \tag{13}$$

is not sufficient for a complete characterization of its second-order statistical structure. Specifically, the cross-spectral matrix is sufficient if and only if

$$\langle z\bar{z}^\dagger \rangle = \begin{pmatrix} \langle x^2 \rangle & \langle xy \rangle \\ \langle xy \rangle & \langle y^2 \rangle \end{pmatrix}, \tag{14}$$

vanishes, in which case $z$ is called *proper*. The matrix $\langle z\bar{z}^\dagger \rangle$ is referred to as the *conjugate covariance matrix*[32,71]. In the context of frequency-domain signal processing, however, covariance and correlation matrices are referred to as cross-spectral and coherence matrices, respectively. To stay consistent with this terminology, we hence will refer to $\langle z\bar{z}^\dagger \rangle$ as the *conjugate cross-spectral matrix*. The conjugate coherence between $x$ and $y$ can be defined as the coherence between $x$ and $\bar{y}$:

$$\rho_{x,\bar{y}} = \frac{\langle xy \rangle}{\sqrt{\langle |x|^2 \rangle \langle |y|^2 \rangle}}. \tag{15}$$

Thus, $z$ is proper if and only if its conjugate coherence matrix vanishes. We note that the conjugate coherence matrix is not an actual coherence matrix in the sense that it is not Hermitian and positive semi-definite. Furthermore, its diagonal entries, which are called *circularity coefficients*, are generally complex-valued and therefore cannot be interpreted as variances. The circularity coefficient of $x$ measures the deviation of the probability distribution of $x$ from being circular, i.e. invariant under phase-rotations $x \mapsto e^{i\phi}x$. In particular, $x$ is proper if and only if its real and imaginary parts have equal variances and are independent[32,71].

**Relation between power correlation and coherence**. Let $x$ and $y$ be two observed zero-mean complex-valued signals. We show that

$$r_{x,y} = \frac{|\rho_{x,y}|^2 + K_{x,y} + |\rho_{x,\bar{y}}|^2}{\sqrt{(1 + K_x + |\rho_{x,\bar{x}}|^2)(1 + K_y + |\rho_{y,\bar{y}}|^2)}}. \tag{16}$$

Writing the numerator of $r$ Eq. (5) in terms of joint moments and cumulants gives

$$\langle (|x|^2 - \langle |x|^2 \rangle)(|y|^2 - \langle |y|^2 \rangle) \rangle = |\mu(x, \bar{y})|^2 + \kappa(x, y, \bar{x}, \bar{y}) + |\mu(x, y)|^2. \tag{17}$$

Setting $y = x$ in Eq. (17) gives the terms inside the square-root of the denominator of $r_{x,y}$ (Eq. (5)):

$$\langle (|x|^2 - \langle |x|^2 \rangle)^2 \rangle = |\mu(x, \bar{x})|^2 + \kappa(x, x, \bar{x}, \bar{x}) + |\mu(x, x)|^2, \tag{18}$$

and

$$\langle (|y|^2 - \langle |y|^2 \rangle)^2 \rangle = |\mu(y, \bar{y})|^2 + \kappa(y, y, \bar{y}, \bar{y}) + |\mu(y, y)|^2. \tag{19}$$

Dividing Eq. (17) by $\sqrt{\langle |x|^2 \rangle \langle |y|^2 \rangle}$ and using the definitions of the coherence (Eq. (4)), cokurtosis (Eq. (12)) and conjugate coherence (Eq. (15)) gives the numerator of Eq. (16). Likewise, dividing Eqs. (18) by $\sqrt{\langle |x|^2 \rangle}$ and Eq. (19) by $\sqrt{\langle |y|^2 \rangle}$ and using the definitions of the kurtosis (Eqs. (10) and (11)) and the conjugate coherence, gives the two terms inside the square-root of the denominator of Eq. (16). This establishes Eq. (16).

**Non-Gaussian power correlation**. We define the non-Gaussian power correlation between two signals $x$ and $y$ as

$$r^e_{x,y} = \frac{K_{x,y}}{\sqrt{(1 + K_x)(1 + K_y)}}. \tag{20}$$

As a measure of functional connectivity, the non-Gaussian power correlation is to be preferred over the cokurtosis $K_{x,y}$. To see why, we first express the kurtosis in terms of the power correlation as

$$K_{x,y} = \sqrt{(1 + K_x)(1 + K_y)} r_{x,y} - |\rho_{x,y}|^2, \tag{21}$$

and note that

$$\mathbb{V}(|x|^2) = \langle (|x|^2 - \langle |x|^2 \rangle)^2 \rangle = 1 + K_x, \tag{22}$$

and

$$\mathbb{V}(|y|^2) = \langle (|y|^2 - \langle |y|^2 \rangle)^2 \rangle = 1 + K_y, \tag{23}$$

where $\mathbb{V}(|x|^2)$ and $\mathbb{V}(|y|^2)$ are the variances of $|x|^2$ and $|y|^2$, respectively. Denoting the covariance between the power of $x$ and $y$ by $\gamma_{|x|^2, |y|^2}$ and noting that

$$\gamma_{|x|^2, |y|^2} = \sqrt{(1 + K_x)(1 + K_y)} r_{x,y}, \tag{24}$$

we obtain the following expression for the cokurtosis:

$$K_{x,y} = \gamma_{|x|^2, |y|^2} - |\rho_{x,y}|^2. \tag{25}$$

It shows that the cokurtosis equals the covariance between the power of $x$ and $y$, relative to that between matched proper Gaussian signals. However, the covariance $\gamma_{|x|^2, |y|^2}$ scales with the variances of the power of $x$ and $y$, which is undesirable. This is similar to the fact that the covariance between two signals scales with the variance between the signals and therefore needs to be normalized by the product of the standard deviations of the signals. Similarly, we normalize the cokurtosis by dividing by the product of the standard deviations of the power of $x$ and $y$. This then yields the non-Gaussian power correlation.

**Relative contributions**. Since MEG signals are proper, the pseudo-coherences vanish for sufficiently long signals and Eq. (16) reduces to

$$r_{x,y} = \frac{|\rho_{x,y}|^2 + K_{x,y}}{\sqrt{(1 + K_x)(1 + K_y)}}. \tag{26}$$

The relative contribution of the (magnitude squared) coherence to the power correlation is defined as

$$c_{x,y} = \frac{|\rho_{x,y}|^2}{|\rho_{x,y}|^2 + K_{x,y}}, \tag{27}$$

and the relative contribution of the cokurtosis as $1 - c_{x,y}$.

**Effect of signal orthogonalization**. Let $x$ and $y$ be two observed complex zero-mean signals. In ref. [11], $y$ is orthogonalized with respect to $x$ by subtracting the instantaneous linear contribution of $x$. This yields an orthogonalized signal $y_\perp = y - \alpha x$, where

$$\alpha = \sqrt{\frac{\langle |y|^2 \rangle}{\langle |x|^2 \rangle}} \text{Re}(\rho_{x,y}). \tag{28}$$

It is constructed in such a way that $x$ and $y_\perp$ have purely imaginary cross-spectrum: $\gamma_{x,y_\perp} = i\text{Im}(\gamma_{x,y})$. The relative contribution of the coherence between $x$ and $y_\perp$ to their power correlation is

$$c_{x,y_\perp} = \frac{|\rho_{x,y_\perp}|^2}{|\rho_{x,y_\perp}|^2 + K_{x,y_\perp}}. \tag{29}$$

The magnitude squared coherence and cokurtosis between $x$ and $y_\perp$ are given by

$$|\rho_{x,y_\perp}|^2 = \frac{\text{Im}(\rho_{x,y})^2}{1 - \text{Re}(\rho_{x,y})^2}, \tag{30}$$

and

$$K_{x,y_\perp} = \frac{K_{x,y} - 2\text{Re}(\tilde{\kappa}(x, y, \bar{x}, \bar{x}))\text{Re}(\rho_{x,y}) + K_x\text{Re}(\rho_{x,y})^2}{1 - \text{Re}(\rho_{x,y})^2}, \tag{31}$$

respectively. The term $\text{Re}(\tilde{\kappa}(x, y, \bar{x}, \bar{x}))$ can be neglected (see SI) hence we obtain

$$c_{x,y_\perp} = \frac{\text{Im}(\rho_{x,y})^2}{\text{Im}(\rho_{x,y})^2 + K_{x,y} + K_x\text{Re}(\rho_{x,y})^2}. \tag{32}$$

Note that if $x$ is Gaussian, i.e. $K_x = 0$, then also $K_{x,y} = 0$, and hence the relative contribution of the coherence becomes 1. If, in addition, $\text{Im}(\rho_{x,y}) = 0$, the contribution of the coherence vanishes. If $x$ is non-Gaussian and $\text{Im}(\rho_{x,y}) \neq 0$, the relative contribution of the coherence can be rewritten as

$$c_{x,y_\perp} = \frac{|\rho_{x,y}|^2}{|\rho_{x,y}|^2 + K_{x,y} + K_x\text{Re}(\rho_{x,y})^2 + \cot^2(\phi_{x,y})(K_{x,y} + K_x\text{Re}(\rho_{x,y})^2)}, \tag{33}$$

where $\phi_{x,y} = \arg(\rho_{x,y})$. Comparing Eq. (33) with the definition of $c_{x,y}$ in Eq. (27), we see that the effect of signal orthogonalization on the relative contribution of the coherence is determined by the term

$$\eta_{x,y} = K_x\text{Re}(\rho_{x,y})^2 + \cot^2(\phi_{x,y})\left(K_{x,y} + K_x\text{Re}(\rho_{x,y})^2\right). \tag{34}$$

Equation (34) provides some insight into the power correlation between orthogonalized signals. First, since in experimental MEG data, $K_x, K_{x,y} \geq 0$, this implies that $\eta_{x,y} \geq 0$ as well, which shows that the contribution of the coherence can decrease but not increase. Second, if the expected phase-difference $\phi_{x,y}$ between $x$ and $y$ is small, the term $\cot^2(\phi_{x,y})$ becomes large hence $c_{x,y_\perp}$ will be small. In particular, $\phi_{x,y} = 0$ implies that $c_{x,y_\perp} = 0$. This shows that if $x$ and $y$ that are coherent with zero lag, the power correlation between $x$ and $y_\perp$ equals the non-Gaussian power correlation $K_{x,y}/\sqrt{(1 + Kx)(1 + Ky)}$ between the $x$ and $y$.

**Co-occurrent bursts**. For each pair $(x, y)$ of source-projected MEG signals from an individual subject, we z-scored $x$ and $y$ by subtracting their means and dividing by their standard-deviations, yielding standardized signals $x'$ and $y'$. Subsequently, $y'$ was orthogonalized to $x'$, yielding $y'_\perp$. Bursts were extracted by binarizing the absolute values of the real parts of the signals $x'$ and $y'_\perp$ at a threshold value of 3.

The imaginary parts could be used instead of the real parts, but give the same results due to the propriety of the signals and the fact the orthogonalization preserves propriety. This yielded binary signals $x''$ and $y''_\perp$ with ones corresponding to bursts and zeros corresponding to no bursts. The co-occurrence of bursts was quantified by the dot product of $x''$ and $y''_\perp$, divided by the length of the signals. Gathering the co-occurrences of all signal pairs yielded subject-specific co-occurence matrices, which were subsequently averaged over subjects. The above procedure was carried out separately in the alpha and beta frequency bands.

**Computational model.** We employed a corticothalamic mean-field model[27,34,72], which describes the aggregate activity of a neuronal population in terms of their firing rate $\phi_a$ and mean membrane potential $V_a$ with $a \in \{e, i, r, s\}$. The corticothalamic mean-field model encompasses two cortical populations (excitatory $e$, inhibitory $i$) and two thalamic populations (relay $s$, reticular $r$). The membrane potential of a population fluctuates $V_a(t)$ as a result of the incoming firing rate $\phi_a(t)$ from other population and/or itself according to

$$\left(\frac{1}{\alpha\beta}\frac{d^2}{dt^2} + \left(\frac{1}{\alpha}+\frac{1}{\beta}\right)\frac{d}{dt} + 1\right)V_a(t) = \sum_{a'}\nu_{aa'}\phi_{a'}(t) + \sum_{b}\nu_{ab}\phi_b(t-\tau). \quad (35)$$

The constants $\alpha$ and $\beta$ refer to the synaptic rise and decay constants, $\nu_{aa'}$ and $\nu_{ab}$ to the connection strength between populations, where $\nu_{ab}$ refers to connections between the thalamic and cortical populations. Propagation between thalamic and cortical populations is delayed by $\tau$. At the cell body, the membrane potential $V_a$ is transformed into a firing rate using a sigmoid function

$$Q_a(t) = \frac{Q_{max}}{1 + \exp\left(-\left(V_a(t)-\theta\right)/\sigma\right)}. \quad (36)$$

The mean firing rate is further temporally damped using the following expression

$$\left(\frac{1}{\gamma_a^2}\frac{d^2}{dt^2} + \frac{1}{\gamma_a}\frac{d}{dt} + 1\right)\phi_a(t) = Q_a(t), \quad (37)$$

with $\gamma_a$ being the temporal damping rate. For inhibitory, relay and reticular populations, $\gamma_a \approx \infty$, hence $\phi_a(t) = Q_a(t)$. Parameter values are exactly the same as in ref. [34].

To analyse power correlations, we coupled two corticothalamic models by connecting their excitatory populations to each other, controlled by a coupling parameter. Similar as in ref. [27], we implemented state-dependent noise to obtain switching in a multistable system between noise-induced fluctuations in the linear regime and limit cycle behaviour close to a Hopf bifurcation (see Hopf bifurcation in Fig. 2b reconstructed by DDE biftool[73]). Hence, there is switching between low amplitude fluctuations and high amplitude oscillations. State-dependent noise was incorporated in the thalamic relay population and added to Eq. (29)[27]

$$\phi_n(t) = \phi_n^{(0)} + \phi_n^{(m)}(t) + \chi\phi_n^{(j)}(t)\phi_e(t-\tau). \quad (38)$$

The first term is a constant reflecting the mean and was set to $\phi_n^{(0)} = 0.5$. The second and third term dependent on random values $\phi_n^{(a)}$ and $\phi_n^{(m)}$ drawn from a Gaussian distribution with standard deviation equal to 0.1. The parameter $\chi$ controls the balance between purely additive noise $\phi_n^{(a)}$ and the multiplicative noise $\phi_n^{(m)}$. To analyse the power correlation and the contribution of coherence and cokurtosis we varied the coupling parameter and $\chi$. All model equations (Eqs. (35)–(38)) were solved in Matlab using the Euler-Maruyama method with sufficiently small time step ($1 \times 10^{-4}$ s), i.e. smaller than the magnitude of the corticothalamic delays.

**Statistics and reproducibility.** All results on the MEG data are validated using non-parametric methods (surrogate data and bootstrapping) and their reproducibility is demonstrated by repeating all analyses on independent scanning sessions.

**Reporting summary.** Further information on research design is available in the Nature Portfolio Reporting Summary linked to this article.

## Data availability
The MEG and anatomical data used in this study have been provided by the Human Connectome Project[65]. The data and documentation can be downloaded from https://www.humanconnectome.org.

## Code availability
All custom code was written in Matlab (MathWorks) Version 2019a (see mathworks.com). Matlab code for source-projection of MEG sensor signals and the calculation of the functional networks is available at https://github.com/Prejaas/amplitudecoupling.

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

## Acknowledgements

R.H. was funded by NWO-Wiskundeclusters grant nr. 613.009.105. P.K.B.T. was funded by an EMBO New Venture Fellowship 9139 and an EAN Research Experience Fellowship awarded to P.K.B.T.

## Author contributions

R.H. and P.K.B.T. designed the study, R.H. carried out the data analyses, P.K.B.T. carried out the computational modelling, R.H. and P.K.B.T. wrote the manuscript.

## Competing interests

The authors declare no competing interests.
