## [Peer Review File · Communications Biology]

Reviewers' comments:

Reviewer #1 (Remarks to the Author):

The paper discusses the mathematical relationship between the coherence and kurtosis measures with power correlation networks in electrophysiological signals. Overall, the paper is very well written and clearly discusses the phase and amplitude coupling in power correlation networks using biophysical simulations and empirical magnetoencephalography (MEG) data. I have a few minor comments and suggestions about the "empirical MEG data" section as follows.

First, the analysis for the co-occurrent bursts in cortical activity is conducted at the group-level. The authors should discuss variability of the burst co-occurrence matrices across subjects and show examples of the burst co-occurrence matrix for individual subjects.

Second, the authors should further discuss the sensitivity of the high amplitude bursts to noise and artefacts in MEG data.

Lastly, the authors should provide the details of the k-means clustering for the cokurtosis matrices. For example, it was not clear how they chose the number of clusters and whether they considered variation across replications.

Reviewer #2 (Remarks to the Author):

This paper proposes an expression relating power correlation and coherence that is exact and holds for any two non-Gaussian signals, with three implications: it explains the common and complementary nature of phase and amplitude coupling; it elucidates the contribution of coincident bursting events to power correlation, thus providing a window to the neuronal origin of amplitude coupling. Based on this, the authors characterise data from simulations and MEG data.

Overall, I think this is a very nice contribution to the field, which extensively interprets these measures without having a clear mathematical understanding of them. In doing that, this paper clarifies a lot of previous work that used these measures. Bringing cokurtosis as an alternative measure of connectivity that brings together the other two, and also has its own benefits, seems also useful and worth being published. Therefore, I only have relatively minor comments, suggestions to improve readability, and a few typos to flag. More or less ordered by appearance in the text:

- In general, there are a lot of people in neuroimaging (ie human fMRI) —and also in electrophysiology— that claims that it's all in the covariance. I understand this paper is contrary to this claim, because had the covariance all the information, it would basically mean that signals are close to Gaussian —but you suggest otherwise . This is worth discussing I think.

- It's obvious for many people but not necessarily for everyone: what is it meant by an electrophysiological signal not being Gaussian? One could naively think that by Gaussian you mean white noise. By Figure 1, this becomes clear, but maybe it could be made more explicit earlier.

- It is claimed that this analysis "hence provides a window to the neuronal origin of amplitude coupling". I guess at some level it does, if amplitude coupling always owes to bursting. But what bursting represents neurally and biophysically I guess depends on the frequencies we are talking about and maybe other factors. Given that, I am not sure that this analysis sheds light on the neuronal origin of power correlations *generally*. This doesn't take any value from this work.

- Page 3, "when the system operates near a subcritical Hopf bifurcation" would sound cryptic to many without further (intuitive) explanation.

- Page 4 "we establish that ongoing cortical signals as measured with MEG have zero conjugate coherence". Sorry if I missed something but this reads misleading, because it suggests you proved this mathematically, whereas it's just an empirical observation in a single data set. Are you sure this generalises?

- Following on the previous comment, it would be nice to give an intuition of what conjugate

coherence means as opposed to coherence.

- In Figure 4, the last network labelled as default mode, doesn't it have too much dorsal prefrontal to be called DMN?

- "Power correlations between orthogonalized signals, therefore, are not completely free of spurious connectivity caused by signals leakage". Some people would say that orthogonalized signals are free of signal leakage by definition. Can the authors provide some intuition of why that's not true?

- "... were characterised by more long distance connections between hemispheres". Can this be quantified by some measure?

- "Note that these obtained networks are very similar to networks that were obtained using a time-delay embedded Hidden Markov model in MEG data". This is interesting because the TDE states are Gaussian by definition and do not capture kurtosis as far as I am concerned. I would have guessed that any information on kurtosis would be captured (if it is captured at all) by state changes. Why the obtained networks look similar to the ones in [56] then?

- "we propose cokurtosis or its normalised version, the non-Gaussian power correlation, as a novel functional connectivity metric". Is cokurtosis really novel, or just not very popular? (until the moment this paper gets published I mean).

- In equation 16 of the methods section there is an epsilon variable that is not defined anywhere. In eq. 1 this seems to be ρ . Is this the same?

- The paper doesn't say much about how they implemented the model or the simulations. It would be good to add some more info in this regard. For example, how did you produce figure 2b? Also, it would be very nice if authors could make some code available.

Typos:

- Page 4 "We will use coherence as proxy for phase coupling" -> "We will use coherence as a proxy for phase coupling"

- Page 5 "A positive value therefore means.." -> "A positive value of r^e therefore means.." (as opposed to just r).

- Sorry if I missed, but is ϕ (y axis of figure 2) defined?

- Page 11: "Empirical"

- Page 18: precisely

- Page 19, "be a reflective of" -> "be reflective of" or "be a reflection of"?

Reviewer #3 (Remarks to the Author):

The authors attempted to investigate the dissociation between phase and power correlation networks in the human brain.

This dissociation, as it was partly shown by previous studies, supported by empirical and simulated MEG recordings presented here is driven by co-occurrent bursts.

What do these power correlation networks reflect?

The authors showed that for two non-Gaussian electrophysiological signals, their power correlation depends on their coherence, co-kurtosis and conjugate-coherence.

Also, only coherence and cokurtosis contribute to power correlation networks in MEG data, while co-kurtosis is less affected by artefactual signal leakage and better mirrors haemodynamic resting-state networks revealed with fMRI.

Both simulations and MEG data showed that co-kurtosis may reflect co-occurrent bursting events.

In summary, their findings untangled the origin of

the complementary nature of power correlation networks to phase coupling networks and suggests that the neuronal origin of resting-state networks is partly reflected in co-occurrent bursts in neuronal activity.

My comments are summarized below:

Major comments:

1) In fig. 3a & b, and fig.5 a & b, you should present the mean and standard deviation of the estimates across the sample.

2) In fig. 3 h,i,j, and also in fig.4, and fig.5c,d,e, you demonstrated the power correlation, coherence and co-kurtosis networks.

A) How did you illustrate a matrix of size ROIs x ROIs, as a vector plotted in a brain sketch?

B) Did you illustrate the strength (sum or every row)?

C) Which threshold did you apply to the data in order to underline the important ROIs?

3) It is very important to show in the main manuscript the difference maps between figure 3 & 5, and also the distribution of these differences to further enhance the effect of orthogonalization across the cortex.

4) Apart from coherence, how does trivial connectivity metrics like PLV or better imaginary part of PLV behave compared to coherence?

5) You applied K-means clustering on the co-kurtosis matrices

A) Did you apply it to the group-averaged matrices?

B) How did you select the number of clusters? You can use the silhouette index.

C) How did you concatenate the findings from multiple runs?

D) Did you use Euclidean distance as a proper metric?

6) You discussed the spatial similarity of identified resting-state networks from cokurtosis matrices with the haemodynamic matrices revealed from BOLD resting-state activity.

A) However, we really don't know the sensitivity of BOLD activity to capture neuronal interactions across the cortex. Please comment on this.

B) You mentioned by the end of the discussion part : Co-occurrence of neuronal bursts revealed canonical haemodynamic resting-state networks.

I don't understand why you insist to connect your findings using MEG modality with findings from resting-state fMRI studies.

Moreover, you didn't analyze any BOLD activity in your study under the same framework. For that reason, you should avoid connecting your findings with the potential findings in the case that you have analysed BOLD activity with the same pipeline. Moreover, you can find published articles where they presented MEG resting-state networks similar to the rs-fMRI brain networks.

Suggestion:

Co-occurrence of neuronal bursts revealed canonical electrophysiological resting-state networks

Minor comments:

1) p.16 : Thus, other explanations for the observed positive cokurtosis kurtosis and cokurtosis than bursting events seem unlikely

The second cokurtosis should be replaced or deleted.

We would like to thank the reviewers for the thorough evaluation of our manuscript. We appreciate the comments that contributed to an improved version of this manuscript. Below you will find our detailed response to the reviewers' comments in blue typeface.

Reviewer 1 (Remarks to the Author):

The paper discusses the mathematical relationship between the coherence and kurtosis measures with power correlation networks in electrophysiological signals. Overall, the paper is very well written and clearly discusses the phase and amplitude coupling in power correlation networks using biophysical simulations and empirical magnetoencephalography (MEG) data. I have a few minor comments and suggestions about the "empirical MEG data" section as follows.

We thank the reviewer's assessment of our work.

First, the analysis for the co-occurrent bursts in cortical activity is conducted at the group-level. The authors should discuss variability of the burst co-occurrence matrices across subjects and show examples of the burst co-occurrence matrix for individual subjects. Second, the authors should further discuss the sensitivity of the high amplitude bursts to noise and artefacts in MEG data.

We would like to stress that we used this point-process or thresholding method merely as an illustration. There are better and more sophisticated ways to estimate high amplitude bursts in MEG data. Examples are Hidden Markov Modelling and the "better oscillation detection" (BOSC) method [1, 2]. These methods are also more robust against noise and artefacts in the MEG data compared to our relative simple point-process analysis and should therefore be addressed for questions about variability across subjects.

We have now added this to the discussion:

The strength of the current analysis is that this very simple method already demonstrates the relationship between coincident bursting events and power correlations. However, more sophisticated analysis tools exist to capture bursting events, such as the Hidden Markov modelling approach that is sensitive to distinct spectral features of bursts [3] and the "better oscillation detection" (BOSC) method, which are presumably less sensitive to noise and artefacts in the data [2]."

Lastly, the authors should provide the details of the k-means clustering for the cokurtosis matrices. For example, it was not clear how they chose the number of clusters and whether they considered

variation across replications.

In the original manuscript, the number of clusters was set by hand. In the revised manuscript, we used a formal criterion (the elbow method), which caused two of the four networks in the alpha frequency band to merge. All other networks remained the same. We added the following description to the text:

”To extract non-Gaussian power correlation subnetworks, we clustered the columns of the subject-averaged cokurtosis matrices using k-means clustering. The number of clusters was determined by the elbow method applied to the average sum of squared within-cluster distances to the cluster centers. In selecting the optimal number of clusters, the number of clusters was allowed to range from one to ten. For each of the ten values, we selected the best cluster out of ten replications with different initial conditions.”

and

”The third network comprises the medial frontal, temporal, and parietal areas of the DMN as well as the auditory network. When the number of clusters is increased from three to four, the auditory networks separates from the DMN.”

Reviewer 2 (Remarks to the Author): This paper proposes an expression relating power correlation and coherence that is exact and holds for any two non-Gaussian signals, with three implications: it explains the common and complementary nature of phase and amplitude coupling; it elucidates the contribution of coincident bursting events to power correlation, thus providing a window to the neuronal origin of amplitude coupling. Based on this, the authors characterise data from simulations and MEG data.

Overall, I think this is a very nice contribution to the field, which extensively interprets these measures without having a clear mathematical understanding of them. In doing that, this paper clarifies a lot of previous work that used these measures. Bringing cokurtosis as an alternative measure of connectivity that brings together the other two, and also has its own benefits, seems also useful and worth being published. Therefore, I only have relatively minor comments, suggestions to improve readability, and a few typos to flag. More or less ordered by appearance in the text:

We thank the reviewer’s assessment of our work.

In general, there are a lot of people in neuroimaging (ie human fMRI) —and also in electrophysiology— that claims that it’s all in the covariance. I understand this paper is contrary to this claim, because had the covariance all the information, it would basically mean that signals are close to Gaussian —but you suggest otherwise . This is worth discussing I think.

This is exactly right. To emphasize this, we added the following text to the fourth paragraph in the Introduction:

”For a pair of zero-mean Gaussian signals, all statistical information is contained in their second-order moments, namely the signals’ variances and their covariance (time-domain) or cross-spectrum (frequency domain) and the Gaussian distribution is the only distribution with this property. Thus, the observed dissociation between coherence and amplitude correlation networks in EEG and MEG data [26,4] implies that second-order moments do not contain all information about the signals. This additional information, however, can only be obtained by considering higher-order moments of EEG/MEG signals.”

It’s obvious for many people but not necessarily for everyone: what is it meant by an electrophysiological signal not being Gaussian? One could naively think that by Gaussian you mean white noise. By Figure 1, this becomes clear, but maybe it could be made more explicit earlier.

To clarify what is mend by a pair of signals being Gaussian, we added the following text to the third paragraph of the Introduction:

”A pair of time-domain signals is referred to as Gaussian if the pairs of samples are drawn from a bivariate Gaussian distribution. In EEG/MEG studies, signals are usually analyzed in the frequency-domain and hence are complex-valued. In this case, a pair of signals is Gaussian if the four-vectors of the real and imaginary parts of the samples are drawn from a four-dimensional Gaussian distribution. Note that, whether or not a pair of signals is Gaussian, does not imply anything about the temporal structure of the signals and, in particular, is unrelated to the signals’ auto- and cross-correlation functions.”

It is claimed that this analysis “hence provides a window to the neuronal origin of amplitude coupling”. I guess at some level it does, if amplitude coupling always owes to bursting. But what bursting represents neurally and biophysically I guess depends on the frequencies we are talking about and maybe other factors. Given that, I am not sure that this analysis sheds light on the neuronal origin of

power correlations *generally*. This doesn't take any value from this work.

We indeed agree with the reviewer to rephrase our claims about "neuronal origin". We have now changed this description into "electrophysiological signal properties" as this refers to the underlying properties of electrophysiological signals that result in amplitude coupling, and does not refer to the biophysical origin of amplitude coupling. This specific part of the text now reads:

"it hence provides insight into the electrophysiological signal properties that result in amplitude coupling."

Page 3, "when the system operates near a subcritical Hopf bifurcation" would sound cryptic to many without further (intuitive) explanation.

We have now changed this part of the text to:

"We first demonstrate the relevance of our mathematical expression in simulations with ground truth when the system operates on the edge of instability (a so-called subcritical Hopf bifurcation),..."

-Page 4 "we establish that ongoing cortical signals as measured with MEG have zero conjugate coherence". Sorry if I missed something but this reads misleading, because it suggests you proved this mathematically, whereas it's just an empirical observation in a single data set. Are you sure this generalises?

We agree with the reviewer that this statement reads misleading. To clarify, the Fourier coefficients of any bivariate stationary stochastic process are proper and in particular have zero conjugate coherence. Thus, ongoing MEG signals have zero conjugate coherence in as far as they are stationary. Besides this theoretical argument, in our study we establish that the observed values of the conjugate coherence are not statistically significant. To clarify this issue, we replaced the cited phase by the following text:

"For simplicity we now assume that the MEG signals are jointly stationary. This assumption is in no way essential, but simplifies our analysis of Eq. (1), because it implies that the conjugate coherence terms vanish [Picinbono, 1994]. We will, however, also establish this for the empirical MEG signals in our dataset."

-Following on the previous comment, it would be nice to give an intuition of what conjugate co-

herence means as opposed to coherence.

We added the following explanation to the text, which hopefully provides some intuition:

"The conjugate coherence of a signal with itself measures the extent to which the distribution of its instantaneous phase $\arg(x)$ deviates from being uniform. In particular, it vanishes if and only if the phase is uniformly distributed or, equivalently, the real and imaginary parts of the signal have the same variance and are uncorrelated. In a similar way, the conjugate coherence between two signals measures the extent to which the distribution of the sum $\arg(x) + \arg(y)$ deviates from being uniform. Details are provided in the Methods section."

- In Figure 4, the last network labelled as default mode, doesn't it have too much dorsal prefrontal to be called DMN?

We agree with the reviewer, this network is more reflective of the salience network. We have changed this accordingly: *"The third network comprises areas in the dorsal and medial pre-frontal cortices and shares similarities with the salience network."*

- "Power correlations between orthogonalized signals, therefore, are not completely free of spurious connectivity caused by signals leakage". Some people would say that orthogonalized signals are free of signal leakage by definition. Can the authors provide some intuition of why that's not true?

Orthogonalized signals are indeed completely free of signal leakage, as their coherence is purely imaginary (per construction), but this does not necessarily imply that the correlation between the signals' power envelopes vanishes. In [4] it is shown that orthogonalization completely suppresses signal leakage only for Gaussian signals. And since MEG signals are non-Gaussian, orthogonalization is expected to be less effective. To clarify this, we added the following sentence to the text:

"This is consistent with the fact that orthogonalization completely suppresses signal leakage only for Gaussian signals [4]."

"... were characterised by more long distance connections between hemispheres". Can this be quantified by some measure?

We believe that the manuscript is already quite lengthy. We do understand the reviewer's position

that such a claim requires some quantification and we therefore have removed this statement.

“Note that these obtained networks are very similar to networks that were obtained using a time-delay embedded Hidden Markov model in MEG data”. This is interesting because the TDE states are Gaussian by definition and do not capture kurtosis as far as I am concerned. I would have guessed that any information on kurtosis would be captured (if it is captured at all) by state changes. Why the obtained networks look similar to the ones in [56] then?

The reason why these networks are so similar is most likely because they arise from the same underlying dynamical process, namely coincident bursting, which is reflected in both the second- and fourth-order statistical structure of the MEG signals. This also explains why the networks observed in our study are so similar to the networks obtained using orthogonalized power correlations. We rephrased the relevant text as follows:

“We note that, besides being very similar to the networks obtained by orthogonalized power correlations, these networks are also very similar to the networks obtained using a time-delay embedded hidden Markov model in MEG data [5]. This can be explained by our hypothesis, since the spatial organization of coincident bursting will be imprinted on both the second- and fourth-order statistical structure of the MEG signals. Whereas orthogonalized power correlations and the hidden Markov model capture the second-order (i.e. Gaussian) structure, the cokurtosis captures the fourth-order (i.e. non-Gaussian) structure. The main advantage of using the cokurtosis is that no correction for signal leakage is needed.”

- “we propose cokurtosis or its normalised version, the non-Gaussian power correlation, as a novel functional connectivity metric”. Is cokurtosis really novel, or just not very popular? (until the moment this paper gets published I mean).

We agree with the reviewer that the cokurtosis is not a new measure and that our contribution is limited to demonstrating its use in the context of functional brain connectivity (although we have not come across the particular way that we normalized the cokurtosis to obtain the non-Gaussian power correlation). We have rephrased the text as follows:

“Taken together, we propose to use the cokurtosis, or its normalised version, the non-Gaussian power correlation, as a functional connectivity metric.”

- In equation 16 of the methods section there is an epsilon variable that is not defined anywhere. In eq. 1 this seems to be ρ . Is this the same?

This should indeed be ρ . Thanks for spotting this.

- The paper doesn't say much about how they implemented the model or the simulations. It would be good to add some more info in this regard. For example, how did you produce figure 2b? Also, it would be very nice if authors could make some code available.

We have added text to the method section to explain this: *All model equations (Eq. 35-38) were solved in Matlab using the Euler-Maruyama method with sufficiently small time step ($1 \times 10^{-4}s$), i.e. smaller than the magnitude of the corticothalamic delays. Code can be found at <https://github.com/Prejaas/amplitudecoupling>.*

We have mentioned in the text about figure 2b: *(see Hopf bifurcation in Figure 2a reconstructed by DDE biftool [6])*

Typos:

- Page 4 “We will use coherence as proxy for phase coupling” – > “We will use coherence as a proxy for phase coupling
- Page 5 “A positive value therefore means..” – > “A positive value of r^e therefore means..” (as opposed to just r).
- Page 11: “Empirical”
- Page 18: precisely
- Page 19, “be a reflective of” – > “be reflective of” or “be a reflection of”?

We thank the reviewer for pointing out these typos. We have now corrected them.

- Sorry if I missed, but is ϕ (y axis of figure 2) defined?

ϕ is indeed defined in the methods. It is the mean firing rate of a population. Its meaning is not defined in the legend. We have now mentioned this in the legend of figure 2 too: *“This model simulates electrophysiological signals generated by an excitatory pyramidal neuronal population in terms of its mean firing rate ϕ_e (Figure 2a)”*

Reviewer 3 (Remarks to the Author): The authors attempted to investigate the dissociation between phase and power correlation networks in the human brain. This dissociation, as it was partly shown by previous studies, supported by empirical and simulated MEG recordings presented here is driven by co-occurrent bursts.

What do these power correlation networks reflect?

The authors showed that for two non-Gaussian electrophysiological signals, their power correlation depends on their coherence, co-kurtosis and conjugate-coherence. Also, only coherence and cokurtosis contribute to power correlation networks in MEG data, while co-kurtosis is less affected by artefactual signal leakage and better mirrors haemodynamic resting-state networks revealed with fMRI.

Both simulations and MEG data showed that co-kurtosis may reflect co-occurrent bursting events.

In summary, their findings untangled the origin of the complementary nature of power correlation networks to phase coupling networks and suggests that the neuronal origin of resting-state networks is partly reflected in co-occurrent bursts in neuronal activity.

We thank the reviewer's assessment of our work.

My comments are summarized below: Major comments:

1) In fig. 3a & b, and fig.5 a & b, you should present the mean and standard deviation of the estimates across the sample.

We now added standard errors to the bar plots in Figure 3 and 5 (and Supplementary Figure 3 and 5) and updated the figure legends. The standard errors of the observed contributions were estimated by bootstrapping over subjects (1000 times). We did not add error bars to the contributions of the randomized data, because these were practically zero (less than 0.001). This is mentioned in the figure legends.

2) In fig. 3 h,i,j, and also in fig.4, and fig.5c,d,e, you demonstrated the power correlation, coherence and co-kurtosis networks. A) How did you illustrate a matrix of size ROIs x ROIs, as a vector plotted in a brain sketch? B) Did you illustrate the strength (sum or every row)?

The reviewer is correct, we have illustrated the average over rows of the connectivity matrix. We added the following text to the figure legends: *"(i.e. the row-averaged cokurtosis matrix)"* and similarly for the power correlation and coherence matrices.

C) Which threshold did you apply to the data in order to underline the important ROIs?

The cortical maps were thresholded at their average values. We have now added the following text to the figure legends: *"For better visibility, the cortical maps were thresholded at their average values."*

3) It is very important to show in the main manuscript the difference maps between figure 3 & 5, and also the distribution of these differences to further enhance the effect of orthogonalization across the cortex.

We have now calculated the difference maps for the power correlation, coherence, and cokurtosis in both the alpha and beta frequency bands. Since they are not essential to the main message, we prefer to place them in the supplementary materials (see Supplementary Figure 5). We refer to them in the main text as follows:

"From this we conclude that signal leakage strongly affects Gaussian power correlations, but not non-Gaussian ones, although the cokurtosis maps are slightly altered by orthogonalization (see Supplementary Figure 5)."

4) Apart from coherence, how do trivial connectivity metrics like PLV or better imaginary part of PLV behave compared to coherence?

The reviewer raises an interesting question, which is whether there also exists a relationship between the power correlation and the (imaginary part of the) phase-locking value. This would be very untrivial as the power envelopes and the phase-locking of two signals can be specified independently from one another. A partial result can be found in [Nolte, 2022], in which Nolte and coworkers demonstrated how the PLV can be expressed in terms of the coherence for bivariate Gaussian signals [Nolte, 2022]. This relationship is already quite complicated and it is unclear how to derive similar relationships for more general signal models. We therefore restricted our analysis to the coherence. We have now mentioned in the discussion:

Lastly, we derived an expression for power correlations in terms of coherence and not directly in terms

of phase-locking as measured by the phase-locking value (PLV). Such a relation only exists if assumptions about the signals are made, because the power envelopes and instantaneous phases of a general pair of signals can be specified independently from each other. Nolte and coworkers demonstrated how the PLV can be expressed in terms of coherence for bivariate Gaussian signals [4]. This relationship, however, is quite complicated and involves infinite power series in the coherence. Deriving similar relations for more general (non-Gaussian) models is therefore a challenging question for future studies.

5) You applied K-means clustering on the co-kurtosis matrices

A) Did you apply it to the group-averaged matrices? *Yes.*

B) How did you select the number of clusters? *You can use the silhouette index. We now used the elbow method, but we would like to thank the reviewer for the suggestion.*

C) How did you concatenate the findings from multiple runs? *The clustering was done on the data from the first run; The data from the second run yielded the same networks (see Supplementary Figure 4).*

D) Did you use Euclidean distance as a proper metric? *Yes.*

We have now described in more detail how the clustering was done (see our response to the last comment of Reviewer 1). In our response, the above four questions from the reviewer are addressed.

6) You discussed the spatial similarity of identified resting-state networks from cokurtosis matrices with the haemodynamic matrices revealed from BOLD resting-state activity.

A) However, we really don't know the sensitivity of BOLD activity to capture neuronal interactions across the cortex. Please comment on this.

The reviewer is indeed correct that the non-neuronal (and therefore also neuronal) contribution to BOLD signals is inhomogeneous across the cortex (please see Kurzwaski et al., 2022 J. of neuroscience). However, there is increasing evidence that co-occurrence of high LFP events is one of the most important contributors to BOLD correlations (see Drew 2019, Zhang et al., 2020 neuroimage, Pais-Roldan et al., 2021 neuroimage). We have now adjusted the corresponding part in the discussion where this was mentioned:

”Even though it has been claimed that not more than 10% of the variance of fMRI BOLD can be explained by underlying neuronal events and the contribution of non-neuronal events is inhomogeneous across the brain [7, 8], recent work in mice suggest that kurtosis of calcium signals mirrors kurtosis in BOLD signals and BOLD fMRI correlations are related to nonstationarity of the calcium signals. Furthermore, there is increasing evidence that co-occurrence of high LFP events is one of the most important contributors to BOLD correlations [9, 10, 11].”

B) You mentioned by the end of the discussion part: Co-occurrence of neuronal bursts revealed canonical haemodynamic resting-state networks.

I don’t understand why you insist to connect your findings using MEG modality with findings from resting-state fMRI studies. Moreover, you didn’t analyze any BOLD activity in your study under the same framework. For that reason, you should avoid connecting your findings with the potential findings in the case that you have analysed BOLD activity with the same pipeline. Moreover, you can find published articles where they presented MEG resting-state networks similar to the rs-fMRI brain networks.

Suggestion: Co-occurrence of neuronal bursts revealed canonical electrophysiological resting-state networks

We slightly disagree with the reviewer. Strictly speaking, we have indeed not analysed BOLD activity using our pipeline and there is indeed no reason why MEG studies should treat fMRI work as ‘ground truth’. However, as the reviewer mentioned, there is ample evidence of similarity in networks between MEG and fMRI data. As mentioned in the answer to the previous comment, previous studies have demonstrated using LFP data or calcium imaging that despite the observation of large contributions of non-neuronal signals to BOLD, high amplitude events at the neuronal level are strong contributors to BOLD correlations [9, 10, 11]. Many studies therefore advocate that it is very likely that BOLD networks may partly be a result of correlated neuronal activity as measured by EEG and MEG [12, 13, 14, 15, 16]. Given this argument, we think it is not inappropriate to speculate about the potential relationship between our results and BOLD networks in the discussion.

To meet the reviewer’s concern we have weakened this claim and reformulated this part in the discussion to clarify that this is speculation: *”Therefore, it is tempting to speculate that these resting-state networks could be shaped by coincident neuronal bursting events.”*

Minor comments: 1) p.16 : "Thus, other explanations for the observed positive cokurtosis kurtosis and cokurtosis than bursting events seem unlikely ." The second cokurtosis should be replaced or deleted.

We have corrected this error.

REVIEWERS' COMMENTS:

Reviewer #2 (Remarks to the Author):

Thank you —all my comments have been satisfactory addressed and I support publication.

Reviewer #3 (Remarks to the Author):

I have no further comments.

The authors answered to all my comments.